# Socioeconomic Background and Self-Reported Sleep Quality in Older Adults during the COVID-19 Pandemic: An Analysis of the English Longitudinal Study of Ageing (ELSA)

**DOI:** 10.3390/ijerph20054534

**Published:** 2023-03-03

**Authors:** Adam N. Collinge, Peter A. Bath

**Affiliations:** 1Information School, University of Sheffield, Sheffield S1 4DP, UK; 2School of Health and Related Research (ScHARR), University of Sheffield, Sheffield S1 4DA, UK

**Keywords:** COVID-19, health disparities, health promotion, socioeconomic background, older adults, sleep quality, mental health, physical health, health behaviors, wellness

## Abstract

The COVID-19 pandemic negatively impacted sleep quality. However, research regarding older adults’ sleep quality during the pandemic has been limited. This study examined the association between socioeconomic background (SEB) and older adults’ sleep quality during the COVID-19 pandemic. Data on 7040 adults aged ≥50 were acquired from a COVID-19 sub-study of the English Longitudinal Study of Ageing (ELSA). SEB was operationalized using educational attainment, previous financial situation, and concern about the future financial situation. Sociodemographic, mental health, physical health, and health behavior variables were included as covariates. Chi-squared tests and binary logistic regression were used to examine associations between SEB and sleep quality. Lower educational attainment and greater financial hardship and concerns were associated with poor sleep quality. The relationship between educational attainment and sleep quality was explained by the financial variables, while the relationship between previous financial difficulties and sleep quality was explained by physical health and health behavior variables. Greater financial concerns about the future, poor mental health, and poor physical health were independent risk factors for poor sleep quality in older adults during the pandemic. Healthcare professionals and service providers should consider these issues when supporting older patients with sleep problems and in promoting health and wellness.

## 1. Introduction

In March 2020, the World Health Organization (WHO) declared the novel coronavirus disease 2019 (COVID-19) outbreak to be a pandemic. Since then, the virus has mutated into several variants that have spread around the world [1]. Common symptoms include fever, sustained coughing, and shortness of breath [2]. The virus can cause long-term fatigue (i.e., ‘long COVID’), injury to key organs, and death [3,4,5].

To date, more than six and a half million people have died as a result of COVID-19 [6]. However, while undoubtedly severe, the consequences of the virus have spread far beyond its physical symptoms. Notably, research has identified a general decline in sleep quality during the pandemic [7,8,9]. This is likely to be due to an increase in psychological challenges attributable to the pandemic [10,11,12,13], which may have disrupted sleep quality [9,14,15].

Broadly, disruptions to sleep quality are a primary concern for both scholars and health clinicians due to the overall importance of sleep. On the one hand, good-quality sleep has been shown to improve cognitive functioning, mood, and mental well-being, alongside allowing for somatic regeneration [16,17,18,19]. Conversely, long-term poor-quality sleep may lead to a variety of negative health outcomes, including dementia, cardiovascular and cerebrovascular complications, depression, and cancer [20,21,22,23,24,25,26].

Given these detrimental outcomes, it has been important to identify those who are most at risk of experiencing poor sleep quality and to understand the factors associated with poor sleep so that interventions can be targeted to those in most need. For example, it is now widely accepted that older adults are more likely to experience poor sleep quality than younger and middle-aged adults [27]. This is predominantly due to the natural compression of circadian rhythm waves in old age, which results in a decrease in peak melatonin production [27]. As melatonin is the key moderator of the sleep–wake cycle [28], this natural biological change means that sleep fragmentation and arousal become more frequent, and sleep duration becomes reduced [29], thereby disrupting sleep quality.

In addition to natural biological changes, it is quite possible that the negative relationship between age and sleep quality has been exacerbated by the COVID-19 pandemic. According to Ring et al. [30], older adults typically have a heightened awareness of death, i.e., adults tend to feel closer to death in the later stages of life. However, this heightened awareness is likely to have been compounded by the fact that adults aged 65 and over have been disproportionately affected by the virus in terms of mortality [31,32]; this has been widely publicized throughout the media [33,34]. Consequently, older adults may be more likely to develop an overwhelming fear of COVID-19 [33,35,36,37,38], otherwise known as ‘coronaphobia’, which can present as anxiety, depression, and loneliness [39,40,41]. Such psychological adversity has been shown to negatively impact sleep quality [14,42,43].

Despite the well-documented links between aging and sleep quality, alongside the likelihood that this negative relationship has been compounded by the pandemic, COVID-related research has, so far, failed to thoroughly examine the relationship between aging and sleep quality within the context of the pandemic. Instead, scholarship has generally focused on students, parents, and healthcare professionals (e.g., [44,45,46]). Although such a focus is undeniably important, it has meant that older adults have been somewhat neglected in the literature. Further, the extant literature focusing on age, sleep, and COVID-19 has tended to treat older adults homogeneously (e.g., [29,47]). This is likely to lead to results that are too general; however, this could be corrected by accounting for further factors.

Alongside age, researchers have identified socioeconomic background (SEB) to be a key contributor to sleep problems and health disparities. In other words, lower SEB has been shown to be associated with sleep disorders, including insomnia, alongside overall poor sleep quality [48,49,50,51]. This is likely to be due to higher stress levels experienced by this demographic group compared with the rest of the population [52]. These stressors predominantly revolve around increased economic burdens [53,54], heightened job insecurities [54], and childhood adversity [55,56], which is likely internalized and persists into adulthood [57,58]. Moreover, studies suggest that adults from lower SEBs are more likely to take part in behaviors that disrupt sleep quality, such as alcohol and tobacco consumption [59,60,61,62,63,64,65].

Importantly, SEB, and any health complications associated with it, persists into old age [66]. This suggests that, when analyzing sleep quality in older adults, heterogeneity can be observed by accounting for SEB. The aim of this study was, therefore, to examine the relationship between SEB and sleep quality in older adults in England, in the context of the COVID-19 pandemic. In doing so, this study is important in that it provides new insights into a previously underexplored area and offers new evidence to the growing discourse of sleep problems during the COVID-19 pandemic. 

## 2. Materials and Methods

### 2.1. Data

The data used in this study were acquired from the English Longitudinal Study of Ageing (ELSA) [67]. Since 2002, the ELSA has interviewed a large representative cohort of adults aged 50 or over living in England at biennial intervals regarding their health, social, and financial circumstances. In addition, ELSA also undertook a COVID-19 sub-study, which collected data on pandemic-related variables alongside the usual health and socioeconomic data. The COVID sub-study had two waves. Data collection for wave 1 took place between June and July 2020, and data collection for wave 2 took place between November and December 2020. This study used the COVID-19 sub-study to assess sleep quality in older adults during the COVID-19 pandemic. Specifically, wave 1 data were used as the dataset and had a higher response rate (n = 7040; 75%) than wave 2 (n = 6794; 72%). The data used in this study are freely available at https://www.elsa-project.ac.uk/ (accessed on 27 March 2022).

### 2.2. Sleep Quality

The dependent variable used in this study was sleep quality, which was assessed using the ELSA variable ‘CvHesleep’. For this interview question, participants were asked how well they had slept over the past month using a five-point Likert scale (1 = ‘excellent’, 2 = ‘very good’, 3 = ‘good’, 4 = ‘fair’, and 5 = ‘poor’). This variable was recoded into a binary format for the analyses. Original responses ranging from ‘excellent’ to ‘good’ were recoded as ‘excellent–good’ (=1), whilst ‘fair’ and ‘poor’ responses were recoded as ‘fair–poor’ (=2). This split was based on the premise that good subjective sleep quality is determined by high satisfaction levels [68,69]. Conversely, ‘fair’ responses suggested that participants slept less than adequately, rather than satisfactorily. Hence, ‘fair’ responses were considered to be more closely related to ‘poor’ sleep quality. 

### 2.3. Socioeconomic Background (SEB)

The primary independent variable to be used in this study was SEB. However, there was no single variable measuring SEB in the ELSA dataset. Instead, SEB was operationalized using three proxy indicators: educational attainment (‘w9edqual’), participants’ self-reported financial situation in the three months before the coronavirus outbreak (‘CvPreFn’), and the extent to which participants were worried about their future financial situation (‘CvFinS_CvFinS1_1’). Financial proxies were used on the premise that adults from lower SEBs tend to accumulate fewer savings over their lifetime, and therefore may be more prone to financial difficulties in old age [70]. To this end, greater exposure to financial difficulties and greater concern over future finances were considered to be indicators of low SEB. Similarly, lower levels of education (i.e., <NVQ3/A-Levels) were considered indicative of lower SEB. Neither of the financial proxies were recoded for the purposes of this study. However, the category ‘foreign/other’ (n = 573; 8.1%) was excluded from the education variable as it gave no insight into the level of qualification, and, therefore, could not be used to determine SEB reliably. The educational attainment variable was coded as follows: 1 = ‘Degree or equivalent’ (reference category), 2 = ‘Higher education below degree’, 3 = ‘NVQ3/A-Level’, 4 = ‘NVQ2/O-Level’, 5 = ‘NVQ1/CSE’, and 6 = ‘No qualifications’. The financial variable representing the participants’ self-reported financial situation in the three months before the coronavirus outbreak was coded as: 1 = ‘Living comfortably’ (reference category), 2 = ‘Doing all right’, 3 = ‘Just about getting by’, 4 = ‘Finding it quite difficult’, and 5 = ‘Finding it very difficult’. The financial variable representing the extent to which participants were worried about their future financial situation was coded as: 1 = ‘Not at all worried’ (reference category), 2 = ‘Not very worried’, 3 = ‘Somewhat worried’, 4 = ‘Very worried’, 5 = ‘Extremely worried’. 

### 2.4. Covariates

Alongside the primary dependent and independent variables, a selection of covariates were also included in the statistical analyses. These were categorized as either additional sociodemographic factors, mental health factors, or physical health and health behavior factors, and are described below.

#### 2.4.1. Sociodemographic

Age, gender, ethnicity, and urbanicity were used as sociodemographic covariates. The gender, ethnicity, and urbanicity variables were originally recorded as binary data, so further recoding was not required. Age was recoded from continuous to ordinal data using the following groups: 50–59 years, 60–69 years, 70–79 years, 80–89 years, and 90+ years. Notably, several participants were recorded as having an age of <50 years. As the required age to participate in ELSA is ≥50 years, these were deemed to be errors and these participants were excluded from the analyses.

#### 2.4.2. Mental Health

Feeling depressed or lonely over the past week, and feeling nervous, anxious, or on edge over the past two weeks were used as mental health covariates. The depression and the loneliness variables were used in their original binary form. The anxiety variable was originally based on a four-point Likert scale (1 = ‘not at all’, 2 = ‘several days’, 3 = ‘more than half the days’, and 4 = ‘nearly every day’). This variable was recoded into a binary format, where ‘not at all’ was recoded as ‘no’ (=2) and all other responses were recoded as ‘yes’ (=1). 

#### 2.4.3. Physical Health and Health Behaviors

Self-reported health over the past month, isolating due to increased risk from coronavirus, alcohol and tobacco consumption, and the presence of a longstanding illness, disability, or infirmity (LSIDI) that limits activities were used as physical health and health behavior covariates. The latter variable was a composite, recoded from whether the participant had an LSIDI and whether the LSIDI limits activities. Participants who reported no LSIDI were coded 0. If participants had a LSIDI that did not limit their activities, the response was coded as 1. If participants had a LSIDI that did limit their activities, the response was coded as 2. The alcohol use variable was also recoded as 1 (‘yes’) if they drank alcohol or 2 (‘no’) if they did not drink or had never drunk alcohol. 

### 2.5. Statistical Analysis

Descriptive statistics were first produced to characterize the study sample in terms of sociodemographic characteristics, mental health characteristics, and physical health and health behavior characteristics. Chi-squared tests for independence were conducted to examine bivariate associations between the independent variables and sleep quality. Finally, binary logistic regression was used to determine whether the covariates had a moderating effect on the relationship between the SEB proxy variables and sleep quality in older adults. In total, four regression models were produced. Model 1 included and compared all three SEB proxy variables simultaneously with sleep quality, after which the covariates were added iteratively. Model 2 also included the sociodemographic data, model 3 also included mental health data, and model 4 also included physical health and health behavior data. *p* values, odds ratios, and 95% confidence intervals were calculated to estimate the increased or decreased risks of variables and categories in relation to sleep quality. Variables that were non-significant in the bivariate analysis were excluded from the multivariate analysis. Alpha was set to ≤0.05 throughout this study. Data management and analysis were conducted using IBM SPSS v28 (IBM, Armonk, NY, USA).

### 2.6. Patient and Public Involvement and Engagement (PPIE) 

The research for this study was presented to four members of a Patient and Public Involvement and Engagement (PPIE) panel as part of a post-graduate training scheme organized by Health Data Research (HDR) UK North. Panel members were offered the opportunity to express potential research avenues for this study and which shape suggestions for future research. 

### 2.7. Ethics 

Ethics approvals for the original and subsequent waves of the English Longitudinal Study of Ageing were obtained from appropriate NHS Research Ethics Committees. The University of Sheffield Research Ethics Committee confirmed that this study involved only existing anonymized data and did not require further ethics approval (ref. 046081).

## 3. Results

### 3.1. Characteristics of the Sample 

Table 1 shows the general characteristics of the study sample. Among the 7040 individuals who took part in this wave of the study, just over half (n = 3726; 52.9%) reported having experienced good quality sleep over the past month. The indicators of socioeconomic background (SEB) were relatively high in the study sample, with 42.6% (n = 2999) of all participants achieving A-Levels or higher, 61.2% (n = 4309) of all participants claiming to have been ‘living comfortably’ in the three months before the COVID-19 pandemic, and 77.7% (n = 5470) of all participants claiming to be ‘not very worried’ or ‘not at all worried’ about their future financial status. 

The most frequently occurring age group was people aged 60–69 (n = 2348; 33.4%); 56.5% of the sample were women (n = 3980); the majority came from a non-ethnic minority background (n = 6726; 95.5%) and lived in an urban location (n = 5110; 72.6%). Generally, the frequency of poor mental health was low, with 17.5% of participants experiencing depression (n = 1232), 17.2% reporting loneliness (n = 1209), and 36.5% experiencing anxiety (n = 2572). Overall, 77.6% (n = 5466) of participants rated their health over the past month as ‘good’ or better, while 23.5% (n = 1656) of the sample were self-isolating due to an increased risk from COVID-19. The majority of participants drank alcohol (n = 4596; 65.3%) and did not smoke (n = 6532; 92.8%), while 69.1% (n = 4871) of participants either had no longstanding illness, disability, or infirmity (LSIDI), or their LSIDI did not limit activities. Three hundred and ninety-nine people (5.7%) had had a COVID-19 test; of these, 34 (0.5%) were waiting for results, while and 23 had tested positive (0.3%). 

### 3.2. Bivariate Associations with Sleep Quality

Table 2 presents the results from the Chi-squared tests, which were used to examine associations between the independent variables and sleep quality. Regarding SEB, all three proxy variables were significantly associated with sleep quality in older adults during the COVID-19 pandemic (educational attainment: *p* = 0.005; previous financial situation: *p* < 0.001; and concern over future finances: *p* < 0.001). In addition to these SEB proxies, the following variables were found to be significantly associated with sleep quality in older adults during the COVID-19 pandemic: age (*p* < 0.001); gender (*p* < 0.001); urbanicity (*p* = 0.004); all mental health covariates (all: *p* < 0.001); and all physical health and health behavior covariates (all: *p* < 0.001).

### 3.3. Multivariable Relationships between SEB and Sleep Quality

Table 3 presents the results from the binary logistic regression analysis, which was used to test associations between the socioeconomic background (SEB) proxy variables and sleep quality, in the unadjusted model (model 1, containing only the three SEB variables) and when adjusting for the sociodemographic (model 2), mental health (model 3), and physical health and health behavior (model 4) covariates.

Educational attainment (*p* = 0.321) was not significantly associated with sleep quality in model 1 and remained non-significant when adjusting for the other covariates (model 2: *p* = 0.551; model 3: *p* = 0.635; and model 4: *p* = 0.694).

Conversely, both financial SEB proxies showed a highly significant association in the unadjusted model (model 1) (both: *p* < 0.001). People who felt that they were doing all right (OR = 1.32; 95% CI = 1.16, 1.51), just about getting by (OR = 1.76; 95% CI = 1.36, 2.29), or finding it difficult financially (OR = 2.50; 95% CI=1.28, 4.88) three months before the coronavirus outbreak were significantly more likely to have poor sleep quality during the pandemic compared with people who were living comfortably. Similarly, people who were not very worried (OR = 1.22; 95% CI = 1.07, 1.39), somewhat worried (OR = 2.14; 95% CI = 1.79, 2.54), very worried (OR = 4.11; 95% CI = 2.70, 6.24), or extremely worried (OR = 7.10; 95% CI = 3.19, 15.83) during the pandemic were significantly more likely to experience poor sleep quality compared with people who were not at all worried in the unadjusted model (model 1).

In model 2, when adjusting for age, gender, and urbanicity, both the financial SEB proxies retained a high level of significance (both: *p* < 0.001). Similarly in model 3, when adjusting for anxiety, depression, and loneliness in addition to the sociodemographic variables, the person’s financial situation three months before the pandemic (*p* = 0.003) and being worried about the future financial situation (*p* < 0.001) were significantly associated with poor sleep quality. When adjusting additionally for physical health and health behavior variables (model 4), being worried about the future financial situation remained significantly associated with poor sleep quality (*p* < 0.001), although the financial situation three months before the pandemic was no longer significant (*p* = 0.879). In this final adjusted model, people who were somewhat worried (OR = 1.46; 95% CI = 1.16, 1.83), very worried (OR = 2.02; 95% CI = 1.15, 3.55), or extremely worried (OR = 8.09; 95% CI = 1.62, 40.30) during the pandemic were significantly more likely to experience poor sleep quality compared with people who were not at all worried.

In the final adjusted model, age group (*p* = 0.008), gender (*p* = 0.004), feeling depressed (*p* < 0.001), lonely (*p* < 0.001), and anxious (*p* < 0.001), and self-rated health (*p* < 0.001) were all independently associated with poor sleep quality.

## 4. Discussion

Using data from a large cohort study of adults aged ≥50, this study aimed to explore whether socioeconomic background (SEB) was associated with sleep quality in older adults during the early stages of the COVID-19 pandemic. Further, it looked to examine the effects of additional covariates on the relationships. 

The bivariate results indicated that educational attainment, the person’s financial situation in the three months before the coronavirus outbreak, and the person’s degree of concern over their future financial situation were all associated with sleep quality. More specifically, lower educational attainment and perceived greater financial hardship and concerns were associated with poor sleep quality. Given that these were considered to be indicators of low SEB, it may be concluded that low SEB is related to poor sleep quality in older adults, which is supported by research conducted prior to the COVID-19 pandemic [71,72,73]. Our results show that these relationships between low SEB and poor sleep quality in older adults were present during the pandemic, although, given the cross-sectional nature of our analyses (as further discussed below), it is not possible to say whether they were caused, or exacerbated, by the COVID-19 pandemic. Although COVID-19 infection has been associated with reduced sleep quality, given the low proportion of people who had received a positive COVID test in this wave (0.3%), any differences in sleep quality or the relationship with SEB, were more likely to have been due to concerns about the virus and the pandemic generally, rather than due to actual infection. 

The three education/finance variables were used as proxies for SEB. It may, therefore, be that lower educational attainment and greater financial difficulties and concerns were in themselves important in determining sleep quality in a general sense, rather than as indicators of SEB. This is supported by the finding that the three SEB proxies had varying levels of significance in the multivariate analyses, which might suggest that they are not necessarily indicative of SEB as a whole, and/or that they had a differential role in relation to sleep quality. The three variables are, therefore, each discussed separately below.

Although educational attainment was significantly associated with sleep quality in the bivariate analyses, the association was not significant in the unadjusted regression model. This suggests that the association between educational attainment and sleep quality was explained by the financial variables, i.e., people with lower educational attainment had associated financial concerns, and that these were more important in predicting sleep problems than the level of education itself. Educational attainment remained non-significant when adjusting for sociodemographic, mental health, and physical health and health behavior variables; in other words, while education is related to sleep quality in older adults, the relationship is moderated by a variety of wider factors that may be more directly associated with sleep quality. This finding contrasts with those from previous studies, which found education to have direct causality with poor health [66,71,74]. However, this is likely to be due to the ages of the participants in this study, i.e., an individual’s highest level of education is typically achieved during early adulthood, meaning that a considerable amount of time is likely to have passed between the completion of education and old age. Consequently, the effects of education on (older) adults’ health become less pronounced over time [75], particularly when compared with more recent factors, such as occupation, access to financial resources, or health problems.

Both the person’s financial situation in the three months before the coronavirus pandemic and the person’s degree of concern about their future financial situation were highly significant in the unadjusted model and when adjusted for sociodemographic characteristics. This suggests that older adults with greater financial difficulties and concerns are likely to experience poor sleep quality regardless of their sociodemographic characteristics, e.g., age and gender.

The relationship between the person’s previous financial situation and sleep quality was partially moderated when adjusting for mental health variables, although the category ‘finding it very difficult’ was non-significant across all models. This may have been because of the relatively small numbers in this category (a possible Type II error). This is supported by the finding that, although the category ‘finding it quite difficult’ was significantly associated with poor sleep quality in models 1 and 2, it became non-significant in model 3 as further variables were included. The other categories (‘just about getting by’ and ‘doing all right’) remained significant until model 4, when the health-related variables were included. At this point, the financial situation three months before the coronavirus outbreak was no longer significant. This may indicate that older adults experiencing financial hardship had more physical and mental health problems that were more important in negatively affecting sleep quality. Notably, studies have shown that older adults facing greater financial strain are typically less able to purchase food, pay bills, and own their accommodation outright [76,77,78], which can lead to heightened stress and anxiety [76,77,79]. Feelings of financial inadequacy compared with peers can also promote general life dissatisfaction, resulting in depression [80]. There is, therefore, evidence in the extant literature to support the idea that greater financial difficulty is associated with poor mental health, which can, in turn, negatively affect sleep quality. However, the present study did not directly test for causality, and it is likely that this observed relationship between poor sleep quality and mental health among older adults experiencing greater financial strain is bidirectional.

The inclusion of physical health and health behavior variables into the logistic regression model fully moderated the relationship between participants’ previous financial circumstances and sleep quality. This may suggest that participants experiencing greater financial strain are likely to also have poor physical health, which, in turn, may disrupt sleep quality. In a recent study, König et al. [81] found that older adults experiencing greater financial difficulties are more likely to delay retirement. However, delayed retirement has been found to be associated with musculoskeletal pain [82,83,84], which negatively impacts sleep quality [85,86]. Thus, causal pathways to explain the results of this study may be inferred from the extant literature. However, as with the mental health variables, this relationship may well be bidirectional, given that direct causality was not tested.

With regard to the person’s degree of concern about their future financial situation, this remained significantly associated with sleep quality across all models, and there was limited evidence of a moderating effect when adjusted for mental health variables (model 3). Previous research has shown that financially related concerns about the future have been linked to depression [87,88] and anxiety [89], both of which likely disrupt sleep quality [90]. However, the results of this study suggest that concern over future finances is an independent risk factor for poor sleep quality, i.e., above and beyond the effects of mental health problems. Similarly, the relationship between financial concerns about the future and poor sleep quality was not moderated by the inclusion of physical health variables, similar to findings reported by Morris et al. [91]. It may therefore be concluded that participants who were greatly concerned about the future experienced poor sleep quality, irrespective of any physical health problems. Again, this situates financial concern about the future as an independent risk factor for poor sleep quality. 

### Limitations

Several limitations became apparent whilst undertaking this research. First, much of the data were susceptible to recall bias [92] and under-evaluation [93,94], given that they were acquired using self-reported measures, e.g., the sleep quality and the perceived health variables. Future research could seek to acquire data on sleep quality in a more objective manner in order to allow for improved accuracy. For example, polysomnography could be used to measure sleep quality [95,96], although this requires greater resources and may reduce response rates. The self-reported health measure we used is also subjective, and although self-reported health has long been established as an independent predictor of various health outcomes (e.g., mortality, health service utilization, and prescribed drug use) in older people [97,98], variables derived from a more detailed past medical history may provide a more nuanced understanding of the moderating effect of previously diagnosed physical and mental health on the relationships with sleep problems observed here. Future research could seek to include data on such diagnoses from patient records.

Second, SEB was constructed using three proxy variables for the purposes of this study. While this provided insight into different components of SEB, it did not allow for holistic inferences regarding SEB to be made. Future research should therefore look to use a single SEB composite variable, for example, through measures such as the indices of multiple deprivation (IMD) scale [99]. This may allow for more conclusive inferences to be drawn regarding the relationship between SEB and sleep quality in older adults during the COVID-19 pandemic. 

Third, the categories ‘BAME’ and ‘non-BAME’ were used in the ELSA ethnicity variable. However, these terms may be perceived as too generalized and may misrepresent the disparities faced by certain ethnic groups [100], such as Roma groups [101]. Accounting for the complexity of ethnicity by either using the terms ‘ethnic minority’ and ‘non-ethnic minority’ or by using a more comprehensive list of ethnicities may allow for more representative research to take place. 

Fourth, the dataset used in this study did not account for the range of components that form sleep quality, including sleep duration, sleep fragmentation, and daytime sleepiness [102]. Adopting a narrower focus might yield further interesting results and may highlight greater sleep disparities as a result of SEB. 

Finally, the two waves of data collection during the COVID-19 pandemic were relatively early on, and the data from COVID Wave 1 analyzed here were collected between June and July 2020, when relatively little was known about the virus and how it was transmitted. It is possible that the associations observed here may have been moderated or exacerbated as the pandemic progressed, or the findings may have been different at other times during the pandemic. For example, the UK came out of the initial lockdown period in June 2020; on the one hand, this might have reduced older people’s anxieties about COVID-19 (because the prevalence and risk of COVID-19 were reducing); on the other hand, the increased social activity and lack of rapid testing and vaccination may have increased people’s fears about being infected. These changes may have had a further impact on people’s sleep quality and on the relationship between SEB and sleep. In the medium term, as the pandemic progressed, the understanding of the virus and its transmission increased and rapid tests and vaccines were developed, which probably helped reduce fears, although this may have been offset by the emergence of variants of the virus, e.g., the Beta and Omicron variants, which may have increased uncertainty and fears. Further research should explore whether the relationship between SEB and sleep quality was moderated as the pandemic progressed, e.g., using the (smaller) ELSA COVID Wave 2 study or using other available datasets. Similarly, a longitudinal analysis of individuals before, during, and following the COVID-19 pandemic would provide insights into both the effect of the pandemic on sleep quality and the extent to which the SEB–sleep quality relationship was moderated or exacerbated during this period.

Despite the limitations discussed above, the study involved a large representative sample of people aged 50 and over, and a number of conclusions can be drawn about this study that may be applicable to older people in other countries.

## 5. Conclusions

This study is important in being the first to examine the association between socioeconomic background (SEB) and sleep quality among older people during the early stages of the COVID-19 pandemic in England (June–July 2020). It has demonstrated that, in relation to sleep quality, SEB may need to be conceptualized in relation to component parts in order to understand the factors associated with poor sleep quality. 

In summary, this study found that lower educational attainment, greater exposure to financial difficulties, and greater concern over future finances were all associated with poor sleep quality in older adults during this phase of the COVID-19 pandemic. However, the relationship between educational attainment and sleep quality was moderated and explained by the person’s degree of concern over their future financial situation. Meanwhile, the relationship between the person’s previous financial situation and sleep quality was moderated by mental health, physical health, and health behavior variables. To this end, it cannot be concluded that low SEB as a whole was a significant risk factor for poor sleep quality in older adults. Instead, greater concern over future finances as a specific component of low SEB, poor mental health, and poor physical health were found to be the biggest risk factors for poor sleep quality in older adults during the COVID-19 pandemic. Future research involving longitudinal analysis should investigate how financial concerns, sleep quality, and the relationship between financial concerns and sleep varied before, during, and after the pandemic.

From a practical perspective, the findings of this study highlight ways in which sleep quality can be better managed in older adults. Principally, any financial concerns should be eased as much as possible. While this would probably be best achieved through state pension reforms, such political change does not offer the most practical approach for clinicians. Instead, the establishment of free financial advice centers for older adults in areas of greater deprivation may help ease anxieties relating to future finances. However, such centers must adopt a person-centered approach, given that older adults tend to be reluctant to discuss personal problems with strangers [103]. Other possible options to ease financial strain during future waves of COVID-19, or during future pandemics, may be to provide temporary financial support to the people in greatest need, e.g., older people on low incomes.

In a similar vein, the provision of free and accessible mental health services for older adults should be made a priority in order to reduce cases of anxiety, depression, and loneliness. Efforts must simultaneously be made to promote a mental health discourse in older adult communities, given that mental health is often a stigmatized or unrecognized concept for many older adults [104]. Finally, an active health literacy program could be developed in order to reduce preventable illnesses and behaviors that impede good-quality sleep, such as alcohol consumption, smoking, and obesity. 

This study has developed important insights that could help healthcare professionals, policymakers, and health service providers to support older people generally, as well as during future waves of the COVID-19 pandemic, and in future pandemics more generally. Healthcare professionals should consider the underlying financial or physical/mental health problems faced by individual patients who seek support for sleep problems: although poor sleep may be the problem being present, other associated factors may require consideration. Policymakers and health service providers may wish to consider interventions to promote health and wellness among older people experiencing sleep problems. 

## Figures and Tables

**Table 1 ijerph-20-04534-t001:** General Characteristics of the Study Sample.

Variable	n	%
Sleep quality		
Good	3726	52.9
Poor	2881	40.9
Total valid responses	6607	93.8
Highest educational attainment		
Degree or equivalent	1422	20.2
Higher education below degree	966	13.7
NVQ3/Advanced(A)-Level	611	8.7
NVQ2/Ordinary(O)-Level	1348	19.1
NVQ1/CSE	190	2.7
No qualifications	984	14.0
Total valid responses	5521	78.4
Financial situation 3 months before coronavirus outbreak		
Living comfortably	4309	61.2
Doing all right	2193	31.2
Just about getting by	450	6.4
Finding it quite difficult	63	0.9
Finding it very difficult	24	0.3
Total valid responses	7039	100.0
How worried about: Your future financial situation		
Not at all worried	2742	38.9
Not very worried	2728	38.8
Somewhat worried	1308	18.6
Very worried	185	2.6
Extremely worried	73	1.0
Total valid responses	7036	99.9
Age		
50–59	1294	18.4
60–69	2348	33.4
70–79	2304	32.7
80–89	904	12.8
90+	102	1.4
Total valid responses	6952	98.8
Gender		
Male	3060	43.5
Female	3980	56.5
Total valid responses	7040	100.0
Ethnicity		
Non-BAME	6726	95.5
BAME	314	4.5
Total valid responses	7040	100.0
Urbanicity		
Urban	5110	72.6
Rural	1925	27.3
Total valid responses	7035	99.9
Felt depressed during the past week		
Yes	1232	17.5
No	5797	82.3
Total valid responses	7029	99.8
Felt lonely during the past week		
Yes	1209	17.2
No	5813	82.6
Total valid responses	7022	99.7
Felt nervous, anxious, or on edge over the past two weeks		
Yes	2572	36.5
No	4445	63.1
Total valid responses	7017	99.7
Self-reported health over the past month		
Excellent	685	9.7
Very good	2312	32.8
Good	2469	35.1
Fair	1254	17.8
Poor	313	4.4
Total valid responses	7033	99.9
Why were you staying at home: I am at an increased risk from coronavirus		
Yes	1656	23.5
No	3819	54.2
Total valid responses	5475	77.8
Do you currently drink alcohol?		
Yes	4596	65.3
No	2442	34.7
Total valid responses	7038	100.0
Do you currently smoke?		
Yes	507	7.2
No	6532	92.8
Total valid responses	7038	100.0
Do you have an LSIDI that limits activities?		
No LSIDI	3333	47.3
LSIDI—does not limit activities	1538	21.8
LSIDI—limits activities	2166	30.8
Total valid responses	7037	100.0

NVQ: National Vocational Qualifications; CSE: Certificate of Secondary Education; BAME: Black and Minority Ethnic; LSIDI: long-standing illness, disability, or infirmity.

**Table 2 ijerph-20-04534-t002:** Chi-squared test results.

Variable	Sleep Quality, n (%)		Test Statistic	*p* Value
	Good	Poor		
Highest educational attainment			7.91 *	0.005
Degree or equivalent	880 (61.9)	542 (38.1)		
Higher education below degree	583 (60.4)	383 (39.6)		
NVQ3/Advanced(A)-Level	354 (58.0)	256 (42.0)		
NVQ2/Ordinary(O)-Level	798 (59.2)	550 (40.8)		
NVQ1/CSE	110 (57.9)	80 (42.1)		
No qualification	553 (56.2)	431 (43.8)		
Financial situation 3 months before the coronavirus outbreak			245.63 *	<0.001
Living comfortably	2805 (65.1)	1502 (34.9)		
Doing all right	1174 (53.6)	1018 (46.4)		
Just about getting by	156 (34.7)	294 (65.3)		
Finding it quite difficult	16 (25.4)	47 (74.6)		
Finding it very difficult	4 (16.7)	20 (83.3)		
How worried about: Your future financial situation			315.28 *	<0.001
Not at all worried	1845 (67.3)	897 (32.7)		
Not very worried	1675 (61.4)	1051 (38.6)		
Somewhat worried	575 (44.0)	732 (56.0)		
Very worried	45 (24.3)	140 (75.7)		
Extremely worried	14 (19.2)	59 (80.8)		
Age			44.20 *	<0.001
50–59	683 (52.9)	609 (47.1)		
60–69	1347 (57.4)	1000 (42.6)		
70–79	1426 (61.9)	878 (38.1)		
80–89	583 (64.5)	321 (35.5)		
90+	69 (67.6)	33 (32.4)		
Gender			63.91 **	<0.001
Male	1970 (64.4)	1088 (35.6)		
Female	2186 (54.9)	1793 (45.1)		
Ethnicity			2.21 **	0.137
Non-BAME	3958 (58.9)	2766 (41.1)		
BAME	198 (63.3)	115 (36.7)		
Urbanicity			8.42 **	0.004
Urban	2963 (58.0)	2144 (42.0)		
Rural	1191 (62)	734 (38.1)		
Felt depressed during the past week			534.53 **	<0.001
Yes	384 (29.6)	865 (70.4)		
No	3789 (65.4)	2008 (34.6)		
Felt lonely during the past week			286.01 **	<0.001
Yes	451 (37.3)	757 (62.7)		
No	3700 (63.7)	2111 (36.3)		
Felt nervous, anxious, or on edge over the past two weeks			605.16 **	<0.001
Yes	1031 (40.1)	1539 (59.9)		
No	3116 (70.1)	1328 (29.9)		
Self-reported health over the past month			1040.89 *	<0.001
Excellent	575 (83.9)	110 (16.1)		
Very good	1739 (75.2)	573 (24.8)		
Good	1409 (57.1)	1060 (42.9)		
Fair	370 (29.5)	883 (70.5)		
Poor	59 (18.8)	254 (81.2)		
Why were you staying at home: I am at an increased risk from coronavirus			33.32 **	<0.001
Yes	2382 (62.4)	1436 (37.6)		
No	894 (54.0)	761 (46.0)		
Do you currently drink alcohol?			24.80 **	<0.001
Yes	2813 (61.2)	1783 (38.8)		
No	1343 (55.0)	1097 (45.0)		
Do you currently smoke?			15.46 **	<0.001
Yes	257 (50.7)	250 (49.3)		
No	3899 (59.7)	2631 (40.3)		
Do you have an LSIDI that limits activities?			180.10 *	<0.001
No LSIDI	2188 (65.6)	1145 (34.4)		
LSIDI—does not limit activities	950 (61.8)	588 (38.2)		
LSIDI—limits activities	1017 (47.0)	1147 (53.0)		

* Chi-squared test for trend; ** with continuity correction. NVQ: National Vocational Qualifications; CSE: Certificate of Secondary Education; BAME: Black and Minority Ethnic; LSIDI: long-standing illness, disability, or infirmity.

**Table 3 ijerph-20-04534-t003:** Results of Logistic Regression Models.

	Model 1		Model 2		Model 3		Model 4	
Variable	OR (95% CI)	*p* Value	OR (95% CI)	*p* Value	OR (95% CI)	*p* Value	OR (95% CI)	*p* Value
Highest educational attainment		0.321		0.551		0.635		0.694
Degree or equivalent	Reference		Reference		Reference		Reference	
Higher ed. Below degree	1.06 (0.89–1.25)	0.534	1.05 (0.88–1.25)	0.581	1.08 (0.90–1.30)	0.383	1.03 (0.84–1.28)	0.762
NVQ3/Advanced(A)-Level	1.09 (0.89–1.33)	0.416	1.03 (0.84–1.26)	0.766	1.12 (0.91–1.39)	0.287	1.14 (0.89–1.46)	0.292
NVQ2/Ordinary(O)-Level	1.06 (0.90–1.24)	0.483	1.00 (0.85–1.18)	0.998	1.04 (0.88–1.23)	0.678	0.97 (0.80–1.19)	0.772
NVQ1/CSE	1.11 (0.81–1.52)	0.515	1.11 (0.80–1.53)	0.533	1.03 (0.73–1.44)	0.884	0.83 (0.55–1.26)	0.375
No qualification	1.23 (1.04–1.46)	0.019	1.17 (0.98–1.41)	0.085	1.18 (0.97–1.42)	0.101	0.95 (0.75–1.20)	0.654
Financial situation 3 months before coronavirus outbreak		<0.001		<0.001		0.002		0.879
Living comfortably	Reference		Reference		Reference		Reference	
Doing all right	1.32 (1.16–1.51)	<0.001	1.31 (1.15–1.50)	<0.001	1.23 (1.07–1.42)	0.004	1.05 (0.88–1.24)	0.587
Just about getting by	1.76 (1.36–2.29)	<0.001	1.80 (1.38–2.35)	<0.001	1.61 (1.22–2.14)	0.001	1.02 (0.72–1.44)	0.930
Finding it quite difficult	2.50 (1.28–4.88)	0.007	2.36 (1.20–4.64)	0.013	1.99 (0.98–4.06)	0.058	1.43 (0.55–3.71)	0.464
Finding it very difficult	2.29 (0.61–8.62)	0.221	1.82 (0.47–6.96)	0.385	1.26 (0.30–5.22)	0.750	0.59 (0.10–3.61)	0.571
How worried about: Your future financial situation		<0.001		<0.001		<0.001		0.001
Not at all worried	Reference		Reference		Reference		Reference	
Not very worried	1.22 (1.07–1.39)	0.003	1.20 (1.05–1.37)	0.008	1.12 (0.97–1.29)	0.113	1.10 (0.94–1.30)	0.246
Somewhat worried	2.14 (1.79–2.54)	<0.001	2.02 (1.69–2.42)	<0.001	1.56 (1.29–1.89)	<0.001	1.46 (1.16–1.83)	0.001
Very worried	4.11 (2.70–6.24)	<0.001	3.75 (2.45–5.75)	<0.001	2.06 (1.31–3.23)	0.002	2.02 (1.15–3.55)	0.014
Extremely worried	7.10 (3.19–15.83)	<0.001	7.43 (3.16–17.50)	<0.001	3.44 (1.42–8.32)	0.006	8.09 (1.62–40.30)	0.011
Age				0.620		0.358		0.008
50–59			Reference		Reference		Reference	
60–69			1.00 (0.84–1.19)	0.983	0.96 (0.80–1.14)	0.614	0.80 (0.65–1.00)	0.046
70–79			0.92 (0.78–1.10)	0.375	0.89 (0.74–1.07)	0.199	0.71 (0.57–0.89)	0.003
80–89			0.91 (0.74–1.13)	0.399	0.89 (0.71–1.11)	0.297	0.63 (0.47–0.83)	0.001
90+			0.76 (0.46–1.26)	0.283	0.62 (0.37–1.05)	0.077	0.46 (0.23–0.91)	0.027
Gender				<0.001		0.033		0.004
Female			Reference		Reference		Reference	
Male			0.72 (0.64–0.81)	<0.001	0.88 (0.78–0.99)	0.033	0.81 (0.70–0.93)	0.004
Urbanicity				0.137		0.745		0.902
Urban			Reference		Reference		Reference	
Rural			0.91 (0.80–1.03)	0.137	0.98 (0.86–1.12)	0.745	0.99 (0.85–1.16)	0.902
Felt depressed during the past week						<0.001		<0.001
No					Reference		Reference	
Yes					2.28 (1.91–2.73)	<0.001	1.67 (1.34–2.06)	<0.001
Felt lonely during the past week						<0.001		<0.001
No					Reference		Reference	
Yes					1.51 (1.28–1.78)	<0.001	1.44 (1.18–1.76)	<0.001
Felt nervous, anxious, or on edge over the past two weeks						<0.001		<0.001
No					Reference		Reference	
Yes					2.16 (1.90–2.47)	<0.001	1.86 (1.59–2.18)	<0.001
Self-reported health over the past month								<0.001
Excellent							Reference	
Very good							1.47 (1.10–1.95)	0.009
Good							2.89 (2.17–3.85)	<0.001
Fair							7.39 (5.28–10.35)	<0.001
Poor							9.93 (5.96–16.56)	<0.001
Why were you staying at home: I am at an increased risk from coronavirus								0.8827
No							Reference	
Yes							1.02 (0.87–1.20)	0.827
Do you currently drink alcohol?								0.196
Yes							Reference	
No							0.90 (0.77–1.06)	0.196
Do you currently smoke?								0.996
No							Reference	
Yes							1.00 (0.75–1.34)	0.996
Do you have an LSIDI that limits activities?								0.523
No LSIDI							Reference	
LSIDI—does not limit activities							1.11 (0.93–1.33)	0.257
LSIDI—limits activities							1.05 (0.88–1.26	0.604

NVQ: National Vocational Qualifications; CSE: Certificate of Secondary Education; LSIDI: long-standing illness, disability, or infirmity.

## Data Availability

The data used in this study are available via the study website at https://www.elsa-project.ac.uk/ (accessed on 27 March 2022).

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
