# Peer review of "Socioeconomic Background and Self-Reported Sleep Quality in Older Adults during the COVID-19 Pandemic: An Analysis of the English Longitudinal Study of Ageing (ELSA)"

_ijerph, 2023, doi:10.3390/ijerph20054534_

Round 1

Reviewer 1 Report (Previous Reviewer 1)

This is a resubmitted manuscript. The considerations pointed out in the previous review were met, improving the quality of the work presented. I congratulate the authors for their excellent work.

Author Response

Reviewer 2 Report (New Reviewer)

-Although authors described that they evaluated the 'sleep quality' and socioeconomic background, authors only evaluated subjectively recalled patients' questionnaires. Therefore, I suggest to change the title and corresponding contents more specifically (For example, 'subjectively measured sleep quality')

-Presence of past medical history was not described in current manuscript. Especially, previously diagnosed sleep related disorders, and mental problems such as depression could affect the result.

Author Response

This manuscript is a resubmission of an earlier submission. The following is a list of the peer review reports and author responses from that submission.

Round 1

Reviewer 1 Report

This is a cohort study, carried out since 2002, whose data come from a sub-study which collected data on variables related to the COVID-19 pandemic. The aim of the study was to examine the relationship between socioeconomic background and sleep quality in older adults in England, in the context of the COVID-19 pandemic. The analysed data were collected between June and July 2020, through interviews. Other socioeconomic factors (age, gender, ethnicity and urbanity), mental factors and physical health and healthy behavior factors were introduced as covariates in the analysis. The authors concluded that greater financial concern for the future, poor mental health, and poor physical health and healthy behavior are important risks for poor sleep quality during the COVID-19 pandemic.

The study presented brings important contributions to the area in the research and teaching scope. I congratulate the authors for their excellent work. The introduction and justification of the study were presented in a clear way and based on current references on the subject. The authors consistently point out the need for this study. The proposed method is adequate to the research objectives and the methodological choice is justified and described by the authors. However, with the aim of making the text clearer to the reader, authors are suggested to clarify in the methods section how the variables related to socioeconomic background were analysed, as was described for the other variables, considering that they are the independent variable of interest in this study. The reader only has partial access to this information when reading the results.

The results are related to the proposed objectives and methods. The following, I have only one consideration. The discussion pointed to important aspects of the results. However, the statement in the second paragraph, on page 12, lines 292 to 294, should be revised, since it is not supported by the results presented and is contradictory with the previous sentence.  

Another issue to be reviewed is the need to emphasize to the reader that the results are data collected in the initial period of the pandemic and that it cannot be established that their findings are attributable to the effects of the same. Although this point was addressed in the limitations of the studies, it is important to review excerpts from the discussion and conclusion that may lead the reader to have this interpretation of the results.

Reviewer 2 Report

The study is quite well done but the findings are far from novel.

Financial concerns, poor mental health and poor physical health are always associated with higher prevalence of insomnia regardless of COVID19 or any other global catastrophe.  This has been well established in any age group.

Reviewer 3 Report

The authors present a study, which explores the influence of socioeconomic background on sleep quality during the COVID-19. The work is novel and its publication is recommended once the observations expressed here have been solved.

 The abstract is too long, it should be a total of about 200 words maximum.

Abbreviations should be explained under the tables.

It should be explained what is the importance of the study, and why did the researchers chose the 1 month period of COVID-19. Would the results be any different if the results were taken in any other period outside of the pandemic? Go deeper in the discussion. Did the COVID-19 pandemic negatively impact sleep quality? Which results in this study prove this sentence of abstract? Perhaps a study on socioeconomic background and the sleep quality was carried out before the COVID-19 breakout. It would be useful to compare to see if the COVID-19 pandemic had an effect on the sleep quality.

Were the subjects included in the study suffering of COVID-19? Part of them probably did, or were they excluded from the study? If they had the infection that could have a big effect on their sleep quality during the time of the study. COVID-19 is known to affect sleep quality. If you’re not in the possession of this data, please discuss it based on other research.

I am looking towards receiving an improved version of this manuscript that addresses all these issues .

Best regards

Round 2

Reviewer 2 Report

My concerns were not addressed.
